# Function and Evolution of Nematode RNAi Pathways

**DOI:** 10.3390/ncrna5010008

**Published:** 2019-01-15

**Authors:** Miguel Vasconcelos Almeida, Miguel A. Andrade-Navarro, René F. Ketting

**Affiliations:** 1Institute of Molecular Biology, Ackermannweg 4, 55128 Mainz, Germany; m.almeida@imb.de (M.V.A.); andrade@uni-mainz.de (M.A.A.-N.); 2Faculty of Biology, Johannes Gutenberg Universität, 55122 Mainz, Germany

**Keywords:** small RNA, piRNA, 22G RNA, 26G RNA, 21U RNA, siRNA, Argonaute, RdRP, Piwi, *C. elegans*, nematode

## Abstract

Selfish genetic elements, like transposable elements or viruses, are a threat to genomic stability. A variety of processes, including small RNA-based RNA interference (RNAi)-like pathways, has evolved to counteract these elements. Amongst these, endogenous small interfering RNA and Piwi-interacting RNA (piRNA) pathways were implicated in silencing selfish genetic elements in a variety of organisms. Nematodes have several incredibly specialized, rapidly evolving endogenous RNAi-like pathways serving such purposes. Here, we review recent research regarding the RNAi-like pathways of *Caenorhabditis elegans* as well as those of other nematodes, to provide an evolutionary perspective. We argue that multiple nematode RNAi-like pathways share piRNA-like properties and together form a broad nematode toolkit that allows for silencing of foreign genetic elements.

## 1. Introduction

At any given point, organisms are engaged in a relentless fight for survival. If a threat comes in the form of a pathogen, organisms fight back using innate or adaptive immune systems. A key feature of immune systems is the ability to recognize the self from the non-self, in order to protect the self while ultimately neutralizing the non-self.

At the nucleic acid level, these non-self invaders take the form of transposable elements (TEs) and viruses. These are considered selfish genetic parasites, since they propagate at the expense of the host, often resulting in a decrease in host fitness [1]. Thus, selfish genetic elements can be highly detrimental to hosts and represent a source of genetic conflict. There is a great diversity of pathways that control these elements, hampering their propagation. For example, in prokaryotes, restriction-modification systems and a myriad of Clustered Regularly Interspaced Short Palindromic Repeats (CRISPR) pathways control invasions by bacteriophages and plasmids [2]. In vertebrates, RIG-1-type I interferon (IFN) pathways are important antiviral surveillance platforms [3]. In tetrapods, Kruppel-associated box (KRAB) zinc-finger proteins are a major class of DNA-binding transcription factors that specifically bind and repress TEs [4]. In addition, small RNA (sRNA)-based RNA interference (RNAi)-like pathways are heavily studied platforms of gene regulation. These pathways exist in all clades of life and constitute an important defensive line against non-self genetic elements [5,6,7,8,9].

It has been hypothesized that protection against invading selfish genetic elements actually stimulated the emergence of increasingly more complex modes of gene regulation in eukaryotes [10]. It is indeed appealing to think of chromatin condensation as a denier of DNA access, of the nucleus as a physical gate, of RNA modifications as self “identity cards.” In this view, perhaps all the steps in the normal life of a gene constitute incredibly complex and often redundant processes, which simultaneously assure bona fide self gene expression and work as hurdles for selfish genetic elements. All such hurdles may have evolved as a result of a constant arms race between non-self genetic elements and their hosts, consistent with the “Red Queen” hypothesis [10,11]. As a consequence of their vanguard position in combating non-self sequences, RNAi-like pathways are evolving very fast [12,13,14,15,16].

sRNAs are bound by and guide Argonaute (AGO) proteins to their targets. The AGO cofactor elicits target gene silencing, either by directly cleaving the target RNA or by recruiting additional repressive factors [5,7,17]. Thus, AGO proteins and their partner sRNAs comprise minimal RNA-induced silencing complexes (RISCs), central to RNAi pathways. AGO protein homologs have been identified in all realms of life from Bacteria and Archaea to humans, suggesting an ancient origin and functional importance [9,17]. Eukaryotic AGOs have been characterized in multiple plants, fungi and animals, where they have been shown to be involved in a plethora of processes [5,6,7]. However, the first functional studies on prokaryotic AGOs have only emerged more recently [9,17].

Eukaryotic AGOs comprise a broad family of proteins that can be subdivided into three clades according to sequence homology: the Ago clade, the Piwi clade and the worm-specific (Wago) clade [17] (Figure 1). Ago clade AGOs are ubiquitously expressed and interact with micro RNAs (miRNAs) or small interfering RNAs (siRNAs), the latter either of endogenous or exogenous origin. Piwi clade AGOs bind to Piwi-interacting RNAs (piRNAs). Wago clade AGOs (WAGOs) interact with secondary siRNAs termed 22G-RNAs and regulate a variety of processes, in some cases redundantly.

The biogenesis of miRNAs and endo-siRNAs involves the cleavage of a double-stranded RNA into shorter duplexes by Dicer proteins, conserved RNase III-related enzymes [7]. The shorter duplexes are bound by Ago clade AGOs, which displace one strand of the duplex while maintaining association with the other strand. The latter, known as the guide strand, will guide the AGO protein to targets with base complementarity. Subsequently, AGO proteins can directly cleave the targets or inhibit their translation [7].

In contrast, given the lack of double-stranded RNA intermediates, piRNAs do not require Dicer enzymes for their biogenesis. piRNAs and Piwi proteins are strongly co-expressed in the animal germline where they control the expression of TEs (Box 1) [18,19]. Moreover, Piwi proteins and piRNAs have additional regulatory functions other than TE control, including somatic roles [20].

The nematode *C. elegans* was the model organism wherein RNAi was first described [21]. Twenty years later, we know that these tiny nematodes have the ability to respond to exogenous dsRNA, as well as several incredibly complex endogenous sRNA pathways. Here, we will review the newest developments regarding biogenesis, evolution and function of endogenous sRNA pathways of *C. elegans*. Also, we will show that different sRNA pathways in *C. elegans* share distinct aspects with the metazoan Piwi/piRNA pathways, making it difficult or sometimes even misleading, to talk about a single nematode piRNA pathway.

## 2. General Aspects of the sRNA Pathways of *C. elegans*

*C. elegans* can initiate RNAi upon acquisition of dsRNA from its surroundings. The fact that exogenous dsRNA can be absorbed so easily from the environment reflects the existence of specialized protein factors that promote its internalization from the gut lumen into the intestinal epithelium [25]. Interestingly, dsRNA internalization was recently visualized in vivo, including the deposition of dsRNA into oocytes and embryos [26]. Another key feature of RNAi in *C. elegans* is that it is systemic, that is, dsRNA spreads throughout the pseudocoelom, potentially initiating an RNAi response in most somatic tissues and germline [25].

In addition, *C. elegans* has a complex endogenous sRNA landscape [27,28]. *C. elegans* expresses several classes of endogenous sRNAs, other than miRNAs: 21U-, 26G- and 22G-RNAs. These distinct sRNA populations can be distinguished by size, 5′ nucleotide bias and 5′-end phosphorylation: 21U-RNAs are 21 nucleotides long and have a 5′ bias for uridine monophosphate; 26G-RNAs are 26 nucleotides long and have a 5′ bias for guanosine monophosphate; and 22G-RNAs are 22 nucleotides in length and have a 5′ bias for guanosine triphosphate [28]. 21U-RNAs are likely produced by RNA Polymerase II (RNA PolII) [29], while 26G- and 22G-RNAs are products of RNA-dependent RNA Polymerase (RdRP) activity [30,31,32,33,34,35,36]. RdRPs synthesize sRNAs from template target RNAs in a non-primed and non-processive manner [34,35,37]. Therefore, all RdRP products have an antisense orientation in respect to the gene feature they map to. The genome of *C. elegans* encodes four RdRP genes: *ego-1* and *rrf-1*, necessary for 22G-RNA biogenesis; *rrf-3*, required for 26G-RNA biogenesis; and *rrf-2* of unknown function. 21U- and 26G-RNAs represent primary sRNA species that dictate downstream production of secondary 22G-RNAs [30,32,36,38,39,40]. Secondary 22G-RNA production is also triggered in response to exogenous dsRNA [34,35]. Hence, several primary sRNA inputs result in 22G-RNA production, largely by a common set of factors [41,42,43,44,45]. Since all sRNA pathways converge on this 22G-RNA nexus, many sRNA pathways are in theory competing for shared factors required for 22G-RNA amplification and silencing [41,46]. Distinct subpopulations of 22G-RNAs can be defined by their AGO cofactors and their targets [42,45].

All the described sRNA species associate with at least one of the 27 AGOs that are encoded in the *C. elegans* genome [45]. These AGOs can have overlapping roles, making it challenging to genetically and biochemically dissect these very intricate pathways. The biogenesis of 21U-, 26G- and 22G-RNA species of *C. elegans*, their cofactors and effector functions will be described below. The biogenesis and function of miRNAs in worms have been reviewed elsewhere [47] and will not be discussed here.

## 3. The 21U-RNA Pathway

21U-RNAs were found to interact with the Piwi clade AGO PRG-1 in the germline and were therefore considered the piRNAs of *C. elegans* [38,39,48,49,50]. *prg-2* is another Piwi gene in *C. elegans* but it is currently annotated as a pseudogene (WormBase release WS267, November 2018, at wormbase.org). *prg-1* mutants have a variety of germline defects but are nevertheless viable [39,48,49]. However, when *prg-1* mutant populations are subjected to consecutive bottlenecks, that is, only a reduced number of individuals are allowed to sire the next generation, animals show a progressive deterioration of germline health throughout generations, the so-called mortal germline phenotype [51].

21U-RNAs can be subdivided into two types: type-I and type-II [28,52,53]. Type-I 21U-RNA precursors are likely transcribed by RNA PolII from discrete autonomous genomic loci with a distinctive 5′ upstream motif [28,29,53,54]. The precursors are approximately 26 nucleotides long, with a conventional 5′ cap but are not poly-adenylated [52]. Although more than 15,000 type-I 21U-RNA loci are present throughout the genome, their distribution seems to be highly concentrated in two ~3 Mb clusters on chromosome IV [28]. There are specialized factors dedicated to 21U-RNA precursor transcription (Figure 2A). An additional complex termed the upstream sequence transcription complex (USTC) recognizes the upstream motif and promotes transcription of 21U-RNA precursors [55]. PRDE-1, TOFU-4/5 and SNPC-4 are part of the USTC and all this factors are individually required for 21U-RNA biogenesis [56,57]. Interestingly, SNPC-4 is also interacting with and regulating the expression of small nuclear RNA (snRNA) genes [57]. It was recently shown that the upstream motif seems to be derived from promoter elements that drive snRNA biogenesis [55,58], consistent with the co-option of SNPC-4 in type-I 21U-RNA biogenesis. Altogether, the participation of these specialized factors, together with an amenable chromatin environment [58] provide a framework of how RNA PolII can synthesize these very short, atypical transcripts.

Type-II 21U-RNAs are derived from capped sRNAs that are transcribed by RNA PolII bidirectionally from transcription start-sites [52]. When YRNT motifs are present near the transcription start site, type-II 21U-RNA precursors can be processed into mature 21U-RNAs. It is unknown whether type-II 21U-RNAs have any biological function or are just a by-product of RNA PolII pausing at promoters.

Following transcription, the precursors are exported out of the nucleus but it is not clear how this is achieved. PID-1 is a factor required for 21U-RNA biogenesis that has predicted nuclear localization and export signals [59]. Although functionality of these localization signals was not experimentally verified, a working model implies that PID-1 may be cycling between the cytoplasm and the nucleus, possibly exporting a 21U-RNA precursor. Next, the 5′ cap is removed and the 5′ and 3′ ends of the precursor 21U-RNA are trimmed. PETISCO, a recently identified protein complex containing PID-1, is required for 21U-RNA biogenesis and has potential to bind the 5′-cap and 5′-phosphate of RNA [60,61] (Figure 2A). Future biochemical studies should determine whether PETISCO is involved in processing the 5′ ends of 21U-RNA precursors. PARN-1, a conserved 3′-to-5′exonuclease, trims the 3′ end of 21U-RNA precursors [62]. Finally, the conserved RNA methyltransferase HENN-1 2′-O-methylates the 3′ ends of 21U-RNAs, likely after loading into PRG-1 [63,64,65]. In absence of HENN-1, 21U-RNAs become destabilized.

PRG-1-RISCs engage target transcripts, both protein-coding genes and TEs, based on relaxed sequence complementarity between the 21U-RNA and the target RNA [38,39,40,66,67]. Binding of PRG-1-RISCs to target RNAs tolerates mismatches in most positions, except for positions 2–8 of the 21U-RNA, similarly to miRNA targeting [68,69]. Such target binding results in a broad potential target range, which implies that PRG-1-RISCs have the potential to scan a large sequence-space, potentially the whole germline transcriptome, for non-self transcripts. It follows that there must be additional antagonistic signals to PRG-1-RISCs, in order to avoid wrongful silencing of bona fide germline transcripts. Indeed, there is a complex interplay of several factors and pathways contributing to the ultimate outcome of expressing or silencing a given transcript (see Section 6).

## 4. The 26G-RNA Pathway

The lowly abundant 26G-RNAs are produced by the so-called ERI complex in the germline (Figure 3). The ERI complex consists of several factors, including the RdRP RRF-3 and DCR-1, the *C. elegans* homolog of Dicer [41,70,71]. The conserved CHHC zinc finger GTSF-1 and the Tudor domain-containing protein ERI-5 form a pre-complex with RRF-3, which is required to bring RRF-3 to the remaining ERI complex members [70,71] (Figure 3). The sequence of biochemical events underlying 26G-RNA biogenesis still has to be determined. Also, it is not known how the ERI complex is recruited to specific target transcripts. A current model of 26G-RNA biogenesis implies RNA synthesis by RRF-3 antisense to a template target RNA, creating a dsRNA intermediate [72]. A blunt end is subsequently formed by an exonuclease, presumably ERI-1. The blunt end creates a substrate for DCR-1 cleavage and selectively stimulates the production of a 26 nucleotide long cleavage product [73]. In this model, RRF-3 would then synthesize another 26G-RNA from the next cytosine available 3′ of the DCR-1 cleavage site, thereby initiating another cycle of biogenesis.

ERI complex mutants (e.g., *rrf-3*, *gtsf-1* and multiple *eri* genes) display a broad range of defects [31,41,70,71,74,75,76]: sperm-derived fertility defects including reduced brood size at 20 °C and sterility at 25 °C, high incidence of males (him) and enhanced RNAi (Eri). The latter phenotype is characterized by hypersensitivity to exogenous RNAi and it is believed to be a reflection of competition between exogenous and endogenous RNAi pathways for shared factors [41,42,45,46].

Two distinct subpopulations of 26G-RNAs are produced in two spatially and temporally distinct stages (Figure 3). One subpopulation of 26G-RNAs binds to the AGOs ALG-3 and ALG-4 (henceforth referred to as ALG-3/4) in the spermatogenic germline in L4 hermaphrodites and in male worms [30,33,77]. The other subpopulation of 26G-RNAs associates with ERGO-1 in the oogenic germline and in embryos [32,33,36,74]. Both subpopulations can trigger downstream production of secondary 22G-RNAs that exert effector functions. It remains to be established whether ALG-3/4 and ERGO-1 have target cleavage activities.

### 4.1. ALG-3/4 Branch 26G-RNAs

ALG-3/4 are two, apparently redundant, AGO paralog genes with high sequence similarity that are expressed specifically during spermatogenesis but are absent from mature sperm. Their loss of function phenocopies the abovementioned sperm-derived temperature-sensitive fertility defects [30]. These defects have been attributed to the inability of sperm to activate, specifically by impeding the formation of its characteristic pseudopods that allow for fertilization [30,77]. Of note, dependence of spermatogenesis on temperature is not specific to *C. elegans*. Instead, it seems to be a recurring phenomenon throughout animal evolution [78]. It will be interesting to know if these temperature effects are more generally linked to sRNA pathways.

ALG-3/4-bound 26G-RNAs target spermatogenesis-specific genes, such as major sperm protein genes, kinases and phosphatases required for spermatogenesis [30,77]. Interestingly, the regulatory effects of ALG-3/4 on their targets seem to be complex and dependent on temperature. At normal temperatures, regulation of ALG-3/4 targets seems to be predominantly repressive [30,79]. However, ALG-3/4 appear to promote gene expression at elevated temperatures, in the male germline [77]. Interestingly, such target regulation by ALG-3/4 was shown to be linked to CSR-1 and may occur on the transcriptional level [77]. The authors proposed a model in which spermatogenic ALG-3/4-RISCs trigger biogenesis of downstream 22G-RNAs that, in turn, bind CSR-1 and are paternally transmitted to the next generation, thereby providing a paternal memory of germline gene expression, essential to prevent infertility at elevated temperatures [77]. The molecular mechanisms underlying the temperature dependence of these fertility defects are still not understood. 

Other than temperature, there are additional predictors of the regulatory outcome of ALG-3/4 targeting. Strikingly, ALG-3/4 branch 26G-RNAs preferentially originate from the 5′ and 3′ terminal regions of target transcripts [30,79]. Individual ALG-3/4 targets can have predominant 26G-RNA peaks on the 5′ or on the 3′ terminal regions. Alternatively, some individual targets may display equally abundant 26G-RNA peaks on the 5′ and 3′ ends. A recent study shows that negatively regulated ALG-3/4 targets seem to have predominant 5′ 26G-RNA targeting [79]. Conversely, predominant 3′ targeting by 26G-RNAs tends to be correlated with weaker silencing or even licensing of gene expression. In accordance, ALG-3/4 targets that have predominant 3′ 26G-RNA targeting were found to be more highly expressed than those targeted predominantly on the 5′. These observations are consistent with a model in which 5′ versus 3′ end targeting by ALG-3/4-RISCs is coupled to the regulatory outcome. Clearly, this understanding is incomplete. For instance, an effect of 3′ UTR length on ALG-3/4-mediated regulation was also found [79]. More studies will be needed to thoroughly dissect the temperature-dependent target regulatory effects of ALG-3/4.

### 4.2. ERGO-1 Branch 26G-RNAs

The AGO protein ERGO-1 is expressed in the first two larval stages, in the adult oogenic germline and in embryos [36,63]. ERGO-1 binds a subpopulation of 26G-RNAs that is distinct from ALG-3/4-bound 26G-RNAs (Figure 3). Unlike ALG-3/4-bound 26G-RNAs, ERGO-1 binds 26G-RNAs that are 3’ 2′-O-methylated by the conserved Hen1 methyltransferase ortholog HENN-1 [36,63,64,65]. This methylation of the ribose ring is required for stability of ERGO-1-class 26G-RNAs and influences the accumulation of downstream 22G-RNA populations [63,64,65]. Eri mutants display defects in the accumulation or function of ERGO-1 branch 26G-RNAs and of their respective secondary 22G-RNAs [33,45,70,74,80].

ERGO-1-RISCs target non-conserved, non-germline specific, repeat-enriched sequences like pseudogenes, lincRNAs and recently duplicated genes, triggering secondary 22G-RNA production in these targets [32,33,36,81,82,83]. A recent study showed that ERGO-1 targets are overall small, poorly conserved genes with few introns [82]. Furthermore, the introns of these genes tend to have poor splicing consensus sequences, in comparison to endogenous genes. The authors went on to show that spliceosomes are enriched on these target transcripts [82]. It was proposed that lack of optimal splicing signals and consequent accumulation of stalled spliceosomes triggers sRNA biogenesis. A similar phenomenon was previously reported in the yeast *Criptococcus neoformans* [84]. It should be noted that *C. neoformans* has nuclear RNAi factors that directly interact with spliceosomes but it is unclear how cytoplasmic ERGO-1 activity would connect to nuclear spliceosomes in *C. elegans*.

ERI-6/7 is a homolog of the MOV10L1 and Armitage helicases, which is required for the accumulation of ERGO-1 branch 26G- and 22G-RNAs but not of ALG-3/4 branch sRNAs [81,85]. In accordance, *eri-6/7* mutants share the Eri phenotype with *ergo-1*, *rrf-3* and other Eri genes [81]. Interestingly, ERI-6/7 was not found to physically associate with RRF-3, GTSF-1, DCR-1, ERI-1 or ERI-5, indicating that this factor is not an integral part of the ERI complex [41,70,71]. Mouse MOV10L1 and *Drosophila* Armitage recruit piRNA precursors to initiate piRNA biogenesis [86,87]. An attractive hypothesis, based on ortholog gene function, would be a role for ERI-6/7 in defining target transcripts upon which the ERI complex can be loaded onto, in order to drive 26G-RNA biogenesis. Artificially tethering ERI-6/7 to non-ERI complex targets and probing for de novo 26G-RNA biogenesis would shed light on this hypothesis.

## 5. 22G-RNA Pathways: A Nexus of Gene Regulation

RNAi pathways in *C. elegans* typically involve a primary sRNA that triggers the biogenesis of secondary 22G-RNAs in a feedforward amplification mechanism. As a result, 22G-RNAs are most abundant and can be subdivided into several subpopulations distinguished by the interacting AGOs and sets of targets.

Several factors have been implicated in secondary 22G-RNA biogenesis downstream of 21U- [38,88,89,90] and 26G-RNAs [32,36,43,44,65] (Figure 2A and Figure 3). The RdRPs RRF-1 and EGO-1 are redundantly required for 21U-RNA-driven 22G-RNA biogenesis [38]. In contrast, only RRF-1 was implicated in 26G-RNA-driven 22G-RNA biogenesis [32,36]. Other factors, assumed to be collectively part of a large accessory complex termed Mutator complex are required for the production of both 22G-RNA subpopulations [38,43,44]. Subsequently, 22G-RNAs can associate with cytoplasmic AGOs, like WAGO-1, that elicit post-transcriptional gene silencing (PTGS) of their targets [42,45]. Alternatively, 22G-RNAs can bind to nuclear AGOs, like HRDE-1 and NRDE-3, that lead to transcriptional gene silencing (TGS) of their targets (Figure 2A and Figure 3) [91,92]. TGS is achieved by the assistance of other NRDE nuclear factors that may affect RNA PolII elongation and recruit histone methyltransferases that deposit H3K9me3 and H3K27me3 [88,89,90,91,93,94,95] (Figure 2A and Figure 3).

21U- and 26G-RNA targets are believed to be silenced both by PTGS and TGS [30,36,41,42,45,83,88,89,90,92] (Figure 3). ERGO-1 couples to the somatic nuclear AGO NRDE-3 [65,92], whereas PRG-1 drives loading of HRDE-1 with 22G-RNAs [88,89,90]. Loss of HRDE-1 leads to a mortal germline phenotype, highlighting the importance of establishing and maintaining gene silencing [91]. Conversely, *nrde-3* mutants do not have a distinctive phenotype. Overall, there seem to be cytoplasmic and nuclear elements in endogenous RNAi-like pathways, enabling them to regulate targets through a combination of PTGS and TGS mechanisms.

In some instances, silencing of transgenes and endogenous genes targeted by the 21U-RNA pathway can become stable, independent of the initial PRG-1 trigger and perpetuated transgenerationally, most likely by a combination of 22G-RNA and histone tail modification signals [88,89,90,96]. This form of stable silencing was termed RNA-induced epigenetic silencing (RNAe). Importantly, HRDE-1, NRDE factors, the HP1 homolog HPL-2 and histone methyltransferases seem to be central to RNAe establishment and maintenance.

Interestingly, targets of 26G-RNAs can also maintain 22G-RNAs in the absence of the primary 26G-RNA trigger, in what may be RNAe-like mechanisms. ALG-3/4 branch 22G-RNAs seem to be partially depleted in adult males in response to *gtsf-1* mutation, while in young adults their levels are unaffected [79]. Furthermore, maternal sRNAs elicit zygotic production of 22G-RNAs that are, in turn, maintained throughout development (see Section 7) [79]. Lastly, although ERGO-1 branch 26G-RNAs are expressed only in oogenesis and in embryos, ERGO-1 targets are still targeted by relatively abundant 22G-RNA populations in adult males, suggesting that secondary sRNAs are maintained in the absence of primary 26G-RNA triggers [79]. NRDE-3 is a good candidate AGO to carry on silencing of ERGO-1 targets in the adult male, however expression and function of NRDE-3 in the male soma were not addressed. Altogether, self-perpetuating 22G-RNA populations in the absence of the primary trigger seem to be a recurring theme in *C. elegans* RNAi-like pathways but the underlying molecular mechanisms are unresolved.

## 6. CSR-1 Pathway and Periodic A_n_/T_n_ Clusters Inhibit PRG-1-Mediated Silencing

Amongst the 27 AGOs of *C. elegans*, CSR-1 is the only WAGO protein required for viability [45,97]. CSR-1 is a very enigmatic AGO in that it seems to promote gene expression [77,97,98,99,100]. CSR-1 is expressed in the germline, where it associates with an abundant subpopulation of 22G-RNAs, produced by the RdRP EGO-1, which targets germline-expressed genes. Mutator genes are dispensable for CSR-1-associated 22G-RNAs [44]. This pathway was proposed to provide a memory of self gene expression in the germline [77,97,98,99,100].

In the *C. elegans* germline, PRG-1-RISCs have been shown to perform a transcriptome-wide surveillance of transcripts for silencing [38,40,68,69], while CSR-1 targets expressed genes and seems to promote their activity [98,99,100]. The current model is that either silencing or licensing of a targeted transcript will depend on the outcome of a balance between PRG-1 and CSR-1 targeting (Figure 2B). If PRG-1 targeting prevails, a transcript is flagged as non-self and targeted for silencing. If CSR-1 targeting is stronger, the transcript is recognized as self and is expressed.

Besides the CSR-1 pathway, another prominent signal has been shown to counteract the germline-wide surveillance activity of PRG-1 and to positively affect gene expression: periodic A_n_/T_n_-clusters (PATCs). PATCs are short AT-rich sequences embedded in introns and UTRs of genes that are correlated with germline expression [101]. Single-copy germline-expressed transgenes with foreign sequences, like *gfp*, tend to be silenced in the germline by PRG-1 [40,88,89,90]. Equivalent transgenes with added PATCs to introns and UTRs can bypass silencing by PRG-1 [69,101] (Figure 2B). Of note, PATCs do not seem to be a universal feature of genomes. The genomic signature of PATCs seems to be largely absent in eukaryotes outside the nematode *Caenorhabditis* group. In sum, PATCs and CSR-1 provide signals that antagonize PRG-1 activity in *Caenorhabditis* animals, thereby protecting germline genes from erroneous silencing.

## 7. Parental Contribution of sRNAs

sRNAs and AGOs seem to be most important for gametogenesis and embryonic development in *C. elegans*. Interestingly, all sRNA classes are parentally contributed to the next generation. 21U- and 22G-RNAs are maternally and paternally deposited in embryos [42,66,89,102,103,104,105,106]. Likewise, 26G-RNAs are maternally and paternally provided [33,36,79,104,107]. Maternal contribution of 26G-RNAs is restricted to the oogenic ERGO-1 branch [33,36,79]. Eri mutants lacking ERGO-1 branch 26G- and 22G-RNAs, display a maternal rescue of the Eri phenotype and ERGO-1 target silencing [79,80]. A recent study elucidated the dynamic interplay between maternal and zygotic ERGO-1 branch sRNA populations in establishing gene silencing of targets throughout development [79]. Maternally inherited 26G-RNAs trigger embryonic biogenesis of zygotic 22G-RNAs, which establish target gene silencing. Curiously, in the absence of maternal 26G-RNAs, zygotic 26G-RNAs can still induce the production of 22G-RNAs, attesting for the robustness of this system.

Parentally provided sRNAs were also shown to play an important role in relation to the function of CSR-1 in protecting against erroneous gene silencing. It was demonstrated that in the simultaneous absence of parental 21U-RNAs and RNAe memory, in the form of 22G-RNAs and/or histone tail modifications, progeny that can produce 22G-RNAs are sterile [66,67]. Further dissection revealed that this synthetic sterility arises because WAGOs typically involved, for example, in TE silencing, such as HRDE-1, start to silence typical CSR-1 targets [66,67]. These studies illustrate that a memory of gene expression is transmitted to the progeny via sRNAs, in order to ensure bona fide gene expression and proper development.

## 8. Cross- and Self-Regulation of RNAi-Like Pathways

RNAi in *C. elegans* typically involves rare primary sRNA triggers that elicit the production of abundant secondary 22G-RNAs in a feedforward manner. Since such feedforward amplification mechanisms can exert a considerable strain on biological systems, there are mechanisms to limit their unbridled continuation. As previously mentioned, the Eri phenotype is likely a reflection of competition of exogenous and endogenous RNAi pathways for shared limiting factors [41,46]. It is plausible that such extensive interactions evolved to limit feedforward amplification of RNAi responses. In support of this, exogenous RNAi was shown to affect endogenous sRNA populations and their inheritance [108]. Over a period of 2–3 generations, the endogenous RNAi machinery counteracts and ultimately dilutes the effect of exogenous RNAi. Also, secondary 22G-RNAs arising from exogenous RNAi do not trigger further RdRP activity [109]. Why are such mechanisms employed to control exogenous RNAi responses, while self-perpetuation of 22G-RNAs against endogenous genetic elements still exist? An attractive hypothesis is that exogenous RNAi is especially limited in order to allow incorporation of new environmental inputs in a rapidly changing environment. A recent observation adds one more regulatory mechanism to AGOs. ALG-3 and ALG-4 were found to bind 26G-RNAs that target and dampen the expression of their own mRNAs [79]. In sum, the extensive AGO repertoire of *C. elegans* and its intricate RNAi pathways cross-regulate and self-regulate in a myriad of ways, allowing robust but finite responses to developmental and environmental cues.

## 9. Overview of RNAi-Like Pathways in Other Nematodes

The phylum Nematoda is a very diverse group that is subdivided into fives clades (designated I–V, see Box 2). Homologous RNAi factors and sRNA populations have been profiled in a few members of each class. Although functional studies are lacking, the available studies already shed some light on the evolution of nematode RNAi factors and sRNA classes. These support the notion that nematode RNAi-like pathways are evolving fast [13].

miRNA factors are ubiquitous in nematodes, contrary to many other factors (Table 1). Indeed, miRNAs and their associated machinery, for example ALG-1-like AGOs and DCR-1 enzymes are well conserved across all nematode clades [110,111,112,113,114,115]. Therefore, the ancestral nematode expressed miRNAs and their ancillary factors.

Clade V nematodes have similar RNAi-like pathways to *C. elegans*. Three additional species of the *Caenorhabditis* genus (*C. briggsae*, *C. remanei* and *C. brenneri*) and *Pristionchus pacificus* had their sRNAs profiled [111,113,115,116]. Excluding miRNAs, these species express 26G-, 22G- and 21U-RNAs with largely identical features as their *C. elegans* counterparts. However, specific individual targets and sRNA sequences are not conserved. This scenario is consistent with fast evolving sRNA pathways targeting foreign, fast evolving sequences. An expected exception are CSR-1 targets, which are bona fide germline expressed genes and therefore show a higher degree of conservation than WAGO targets across *Caenorhabditis* [113,116].

Interestingly, some associations have been established between RNAi biology and reproductive life-style. *C. remanei* and *C. brenneri*, which are both gonochoristic, have 2–3 times more 21U-RNAs than the hermaphroditic *C. elegans* and *C. briggsae* [113]. Moreover, the genomes of *C. remanei* and *C. briggsae* encode a total of 37 and 46 AGOs, respectively, almost doubling those of *C. elegans* and *C. briggsae*. The authors proposed that such expansion may have arisen to counteract increased exposure to more diverse TEs, brought about by obligatory mating [113], consistent with a “Red Queen” arms race scenario.

AGO proteins are perhaps the set of factors showing higher plasticity [117]. Although other nematodes have a high number of genomically encoded AGO proteins similar to *C. elegans*, it is often hard to find 1-to-1 orthologues, especially within the WAGO clade [111,114]. Clear ALG-3-like orthologues can be found across all clades, whereas ERGO-1-like AGOs seem to be restricted to clade V (Table 1) [112,117]. Notably, PRG-1-like Piwi AGOs, HENN-1 enzymes, 21U-RNAs and upstream motifs are largely absent outside clade V [111,112,114]. However, other sRNA species were found to target TEs in clades I–IV: the 5′ monophosphorylated 23–25 nucleotide long sRNAs of clade I *Trichinella spiralis* [112]; the 22G-RNAs of clade IV *Globodera pallida* and clade III *Brugia malayi* [112]; and 27 nucleotide long sRNAs with a bias for 5′ triphosphate guanosine or adenosine (or 27GA-RNAs), expressed in three species belonging to the clade IV Strongyloididae family, a particularly relevant group of animal parasites [111].

RRF-3-like RdRP orthologues are distributed throughout the nematode phylum, suggesting that ancestral nematode sRNA biogenesis was perhaps similar to RRF-3-mediated 26G-RNA biogenesis in *C. elegans* (Table 1) [112]. However, sRNA species clearly analogous to *C. elegans* 5′-monophosphorylated 26G-RNAs were not detected outside clade V [111,112], with the exception of the 26G-RNAs of clade III *Ascaris suum* [114]. The 26G-RNAs of *A. suum* are particularly abundant in testes and target spermatogenesis-specific genes, much like ALG-3/4 branch 26G-RNAs in *C. elegans*. Accordingly, three ALG-3/4-like AGOs were identified in *A. suum* [114]. Contrary to *C. elegans*, the *A. suum* spermatogenesis-enriched 26G-RNAs do not show the overall biased distribution toward the 5′ and 3′ ends of target transcripts [30,79]. Altogether, these observations suggest high plasticity of sRNA biogenesis mechanisms by RRF-3-like RdRPs.

The RRF-1 and EGO-1 family of RdRPs is a novelty of clades III-V, indicating that 5′-triphosphorylated sRNAs are not expressed in clades I-II (Table 1) [110,112]. Indeed, 5′-triphosphorylated sRNAs have not been detected in clade I *T. spiralis* and in clade II *Enopolus brevis* and *Odonotophora rectangular* [112]. Absence of 5′-triphosphorylated sRNAs is mirrored by lack of NRDE-3 and HRDE-1 AGOs [110,112,116], implying absence of AGO-driven TGS or the existence of other alternative TGS mechanisms in these nematodes. Supporting the latter, RNA-directed DNA methylation was detected in clade I *T. spiralis* [118] and a strong correlation was observed between methylated regions and sRNA abundance [112]. This may hint at an ancestral nematode mechanism of RNA-directed DNA methylation [112], reminiscent of what is observed in plants [5].

CSR-1 homologs have been identified in clades III and V (Table 1) [110,116]. The presence of CSR-1 genes in clade III is enigmatic because these species lack PRG-1 homologs. It will be interesting to determine whether clade III CSR-1 genes display a positive gene regulatory function in the absence of PRG-1.

## 10. Parallels between Nematode RNAi-Like Pathways and Metazoan piRNAs

The studies referred above illustrate a high level of plasticity of RNAi-like pathways in Nematoda. The putative common ancestor to all nematodes likely expressed miRNAs and a cognate ALG-1/2-like AGO, DCR-1, Piwi AGO(s), endogenous sRNAs made by an RRF-3-like RdRP and RNA-directed heterochromatin formation and DNA methylation [112]. As a result of broad loss of Piwi genes but overall AGO family expansion, we argue that multiple nematode AGOs and sRNA species adopted piRNA-like features. We will substantiate this claim by drawing parallels between nematode RNAi pathways and metazoan Piwi/piRNA pathways, both in mechanistic and functional terms, highlighting common ground and dissimilarities.

### 10.1. Germline Expression

Metazoan Piwi-RISCs are highly expressed in germ cells [8,18]. Hence, germline-specificity could be an argument for defining the Piwi pathway of *C. elegans*. Several findings argue against such simple classification.

First, many *C. elegans* pathways appear to be highly germ cell specific (Table 2). Indeed, 21U-RNAs and PRG-1 are highly enriched in the germline and are required for normal fertility but also ALG-3/4, ERGO-1, their cognate 26G-RNAs, as well as ERI complex factors, are strongly enriched in the spermatogenic and oogenic germlines, respectively [30,33,36,70]. Second, a number of WAGO proteins has been shown to be specifically expressed in germ cells (e.g., HRDE-1, WAGO-1 and WAGO-4) [42,91,106,119]. Third, in further support of the idea that tissue specificity does not provide a handle for defining a Piwi pathway, somatic Piwi/piRNA expression has been convincingly demonstrated in many arthropods and mollusks [120,121]. In fact, such studies hint that the ancestral metazoan used somatic Piwi/piRNA pathways that became increasingly compartmentalized in the germline. We conclude that tissue specificity does not provide any support for classifying a particular *C. elegans* RNAi-like pathway as “the” Piwi/piRNApathway.

### 10.2. Function

Piwi/piRNA pathways comprise specialized RNAi-like pathways that recognize and silence the non-self, most notably TEs (Box 1) [8,18,19]. In this light, piRNA pathways can be thought of as immune systems, given their ability to recognize the non-self, initiate a response, amplify the response and keep memory of the contact, thereby safeguarding future contacts. However, this aspect does not provide a foothold to unambiguously define a *C. elegans* piRNA pathway, as both 21U- and 26G-RNA pathways target distinct sets of non-self transcripts and display these recognition/amplification/memory features.

Mutation of Piwi in flies and mice leads to a range of gametogenesis defects that results in sterility [8,18]. *prg-1* mutant worms are viable but have reduced brood sizes [39,48,50]. As mentioned above, PRG-1 orthologs and 21U-RNAs have been lost in multiple nematode lineages [111,112,114]. These observations suggest that PRG-1/21U-RNA pathways may be less essential to viability than other piRNA systems (Table 2). In contrast, the strongly conserved ALG-3/4-like AGOs are required for normal fertility in *C. elegans* [30]. Therefore, no *C. elegans* RNAi pathway strictly displays complete sterility (except for the CSR-1 pathway), again blurring a parallel with other metazoan piRNA pathways.

Lastly, piRNAs are required in embryos to prime gene silencing that is maintained until adulthood [122]. Similarly, the embryonic activity of both 21U-RNAs and 26G-RNAs seems to be required during embryogenesis [66,67,79,89]. When taking all these arguments into account, no single nematode RNAi pathway is a good direct counterpart of metazoan Piwi/piRNA pathways.

### 10.3. sRNA Features

Are there sRNA features that can help us define “the” worm piRNA pathway? piRNAs have a distinct 5′ uridine bias which is shared by 21U-RNAs [8,18,28,39,48]. Indeed, PRG-1-bound sRNAs have a very strong 5′ uridine bias. However, the length profile of 21U-RNAs, which are almost uniquely 21 nucleotides long, differs from typical piRNAs. The latter, mostly span from 23–35 nucleotides and the populations show a characteristic bell-shaped length distribution (Table 2). 26G-RNAs have a more piRNA-like length distribution but display a different 5′ nucleotide bias. Moreover, 26G-RNAs are produced by an RdRP [30,33,36], not by RNA PolII as typical piRNA precursors [8,18]. Metazoan piRNAs are commonly not defined by individual promoters, a feature shared by 26G- and 22G-RNAs. In contrast, type-I 21U-RNAs are individually transcribed from their own upstream motif. Lastly, piRNAs are 3′ 2′-O-methylated by Hen1 proteins (Box 1). Again, this is not a feature unique to one *C. elegans* pathway, as both 21U-RNAs and ERGO-1 class 26G-RNAs are similarly methylated [63,64,65]. Overall, a single worm pathway homologous to other metazoan piRNA pathways cannot be identified based on sRNA features.

### 10.4. Sequence Homology of AGO Proteins

21U-RNAs were initially classified as piRNAs precisely because of their interaction with PRG-1 [39,48,50]. PRG-1 is without a doubt most closely related to other metazoan Piwi clade AGOs. In addition, ERGO-1 was previously assigned to the Piwi clade [45,63,65]. Our analysis of full length AGO proteins, rooted with two exemplary prokaryotic AGOs, suggests that ERGO-1 is still part of the Ago clade, although very basal (Figure 1). AGOs have several distinct domains with defined functions: the MID domain, which binds the 5′ extremity of the sRNA; the PAZ domain, which accommodates the 3′ end of the sRNA; and the PIWI domain, which is the catalytic domain of AGOs that mediates cleavage. To understand the phylogenetic relationship of ERGO-1 with the eukaryotic AGO clades, we performed phylogenetic analysis of the MID, PAZ and PIWI domains of AGO proteins.

Interestingly, this analysis shows that the MID domain of ERGO-1 does not cluster with any of the eukaryotic AGO clades (Figure 4A). We could not detect homologs for this region of ERGO-1 outside *Caenorhabditis* even using iterative searches (neither with PSIBLAST nor with HMMER). In fact, although ERGO-1 has to retain the ability to bind the 5′ end of the sRNA, its MID domain seems to be quite degenerate (data not shown). The PAZ domain of ERGO-1 is more closely related to Wago clade AGOs than with the Piwi clade, even though it has to accommodate 3′ 2′-O-methylated sRNAs, much like the PAZ domain of Piwi AGOs (Figure 4B). Moreover, the PIWI domain of ERGO-1 is strongly related to that of Ago clade AGOs (Figure 4C). Our phylogenetic analysis does not support previous classifications of ERGO-1 as a Piwi protein. We argue that ERGO-1 is part of the Ago clade and has a Wago-like PAZ domain. Hence, given that the domains of AGO proteins have non-overlapping functions, establishing phylogenetic relationships based on full sequence alignments may mask domain-specific information. Future AGO phylogenetic studies should therefore analyze the different domains separately.

### 10.5. Evolutionarily Conserved Cofactors

Another argument that could be used to define one *C. elegans* RNAi-related pathway as the main equivalent to a metazoan piRNA pathway is the existence of shared cofactors. However, as we will show below, many examples exist that argue against this possibility (Table 2).

Members of the Gametocyte-specific factor 1 (Gtsf1) protein family were found to interact with Piwi proteins in flies [123,124] and mice [125,126]. *C. elegans* has one Gtsf1 ortholog which does not interact with PRG-1 and 21U-RNAs (Table 2). Instead, GTSF-1 was co-opted in *C. elegans* for the biogenesis of both classes of 26G-RNAs, by interacting with the RdRP RRF-3 [70].

A similar scenario holds for the *C. elegans* helicase ERI-6/7, which is required for the accumulation of ERGO-1-class 26G-RNAs (Table 2) [81,85]. Its homologs in flies and mouse, Armitage and MOV10L1, respectively, are piRNA pathway factors [86,87].

Finally, metazoan piRNA pathways typically involve proteins with Tudor domains [7,8]. In *C. elegans*, two Tudor domain-containing proteins, ERI-5 and EKL-1 were implicated in 26G-RNA biogenesis and CSR-1 function, respectively [71,97]. In contrast, no Tudor domain protein has yet been described to act in the PRG-1 pathway.

Similar to the other discussed features, conservation of cofactors does not help to define a single sRNA pathway of *C. elegans* as the main counterpart of metazoan piRNA pathways. Instead, different aspects are represented in different pathways, although most of these tend to cluster in the 21U- and 26G-RNA pathways.

## 11. Concluding Remarks

Historically, an evident first step in understanding nature is to categorize organisms, genes and pathways into distinct classes. While such classification is certainly very useful, it can also be artificial and not fully reflect evolutionary trajectory. We reason that nematode RNAi pathways have blurred borders separating what we define as piRNAs and siRNAs in other metazoans. We believe that referring exclusively to 21U-RNAs as piRNAs is too simplistic and that it may in some cases even be misleading. We propose using the already existing nematode-specific nomenclature to indicate specific sRNA pathways in worms: 21U-, 26G- and 22G-RNA pathways and to refrain from talking about a *C. elegans* piRNA pathway. In fact, a less general terminology overall may be called for in piRNA studies, since numerous differences exist between fly and mouse piRNA pathways, making it dangerous to promptly generalize results.

RNAi-like pathways are broadly used by animals, plants and fungi in the de-escalation of genetic conflict between hosts and non-self sequences. As a result of this relentless arms race, RNAi-like pathways are evolving fast. The existence of several species- or genus-specific RNAi factors and RNAi factor interactions supports such view. The extremely diverse phylum Nematoda epitomizes this evolutionary fluidity, with its many AGOs and sRNA classes interacting in complex ways, both inside and outside the germline, to regulate gene expression in a variety of processes, including gametogenesis and embryogenesis. Further studies on non-model nematodes, including parasitic species of medical and veterinary relevance, are required to elucidate more aspects on the biology of these species, including how RNAi pathways have been and are being evolved.

Box 1Metazoan piRNA pathways.In animals, Piwi clade AGOs and piRNAs are co-expressed and interact predominantly in the germline [8,18]. Their “canonical” function is to protect the germline from the detrimental effects of TEs [8,18,19]. The germline can be considered as an “immortal” tissue, since it continuously gives rise to the next generation. Hence, piRNAs have the important function of preventing uncontrolled TE mobilization from wreaking havoc in the genomes of following generations. Their importance can be attested by the sterility or broad range of fertility defects displayed by piRNA-defective mutants, such as Piwi mutants [8,18]. piRNA biogenesis and function are intensively studied in the germlines of *Drosophila*, silkworm and mouse. Although there are several organism-specific nuances and exceptions, several universal principles of piRNA biogenesis and function can be extracted:
Long piRNA precursor transcripts are typically transcribed by RNA PolII from large genomic regions known as piRNA clusters [8,18]. These clusters harbor degenerate TE copies, relics of old TE invasions. piRNA clusters are considered TE traps, in that once a TE jumps in a cluster at random, complementary piRNAs are produced and this will lead to the silencing of other homolog TE copies;After export of piRNA precursors to the cytoplasm, piRNA production starts by cleavage of piRNA precursors by Piwi AGOs and specialized endo- and exonucleases [18,127]. piRNA maturation is completed when Hen1 enzymes 2′-*O*-methylate the 3′ end of the piRNA. This modification is thought to provide stability to sRNAs;The so-called “ping-pong” amplification cycle involves typically two relaying Piwi AGOs [8,18]. The catalytic activity of one Piwi generates a piRNA that is accepted by another Piwi and this event is repeated in a loop. This feedforward loop allows for robust amplification of the piRNA pool and faithful silencing. The “ping-pong” amplification cycle seems to be an evolutionarily conserved mechanism in Piwi/piRNA pathways [127];Mature piRNAs are generally 23–35nucleotides in length and some have a 5′ bias for uridine [8,18];Piwi-RISCs silence their targets both by post-transcriptional gene silencing (PTGS) and transcriptional gene silencing (TGS) mechanisms [8,18,128]. PTGS is mainly dependent on target cleavage by Piwi AGOs, whereas TGS involves at least one Piwi AGO that is shuttled to the nucleus to target nascent RNAs. Nuclear Piwi AGOs are not sufficient for TGS. Interactions with other factors, such as histone methyltransferases, are required to establish repressive chromatin at target loci;Piwi/piRNA pathways function as an adaptive immune system against genetic parasites. Several features of adaptive immune systems are shared by Piwi/piRNA pathways, like the ability to recognize the threat, initiate a response, amplify the response and keep a memory of the response for further encounters. Memory of past encounters is embedded in piRNA clusters and is thus transmitted to the next generation. In addition, Piwi-RISCs may be directly inherited by the progeny in order to jump-start piRNA biogenesis in the next generation.Several lines of evidence suggest that Piwi/piRNA pathways and TEs are engaged in an evolutionary arms race, consistent with the “Red Queen” hypothesis [13,14,15,16]. Initially developed for host-parasite interactions, this theory may be applicable in the nucleic acid world to Piwi/piRNA pathways and TEs as genetic parasites. In this light, genetic changes beneficial to TEs are counteracted by genetic changes in piRNA pathway factors that eliminate or attenuate the TE advantage. In fact, the *Drosophila* genus seems to be rich in examples supporting an arms race between hosts and TEs [14,15,16]. Also, many factors involved in Piwi/piRNA pathways in diverse organisms are not evolutionarily conserved, suggesting that these pathways are evolving fast, potentially in response to TEs [13].

Box 2Nematode diversity.Nematodes comprise an extremely diverse group of animals with approximately 23,000 described species and an estimated number of more than 1 million of total species [129]. Nematodes can be found in terrestrial or aquatic environments, can be free-living or parasitic to plants and animals and display different modes of reproduction, including gonochorism, hermaphroditism and parthenogenesis. Given this astonishing diversity of life styles, one can easily apprehend that *C. elegans* biology is by no means representative of the whole phylum Nematoda.Phylogenetic analysis have led to the division of the Nematoda phylum into three classes: Dorylaimia (clade I), Enoplia (clade II) and Chromadoria (clades III-V) [129]. The phylogenetic relationship of these classes is still under debate, as their branching order remains unclear. The *Caenorhabditis* genus is part of the Chromadoria, clade V group of nematodes.

## Figures and Tables

**Figure 1 ncrna-05-00008-f001:**
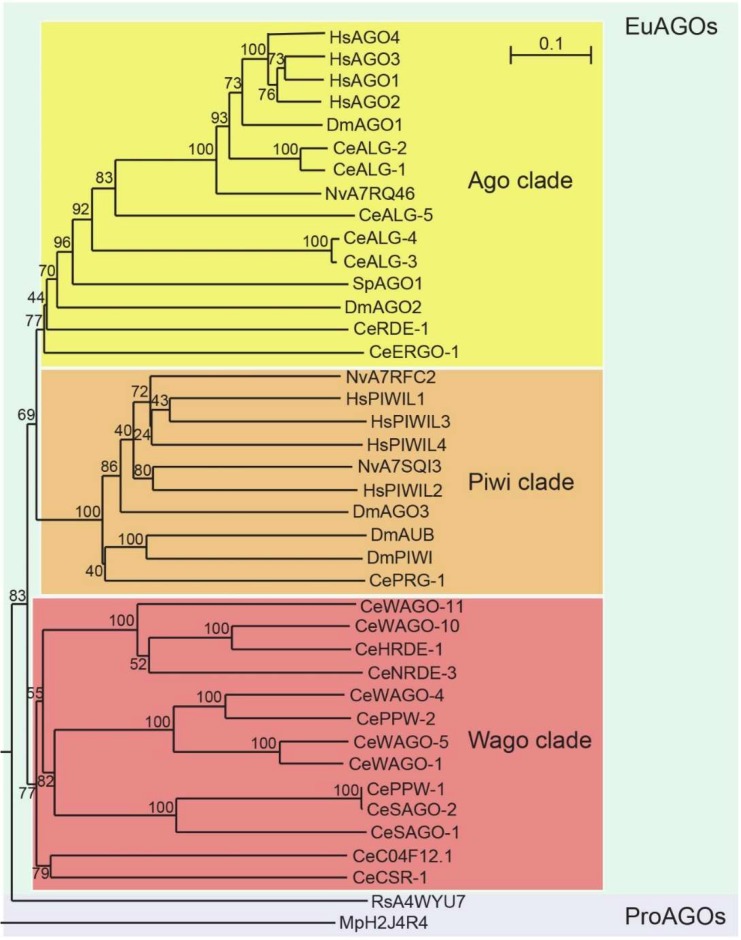
Phylogenetic tree depicting the three main eukaryotic AGO clades. To provide a broad phylogenetic representation across eukaryotes, the AGO proteins of *C. elegans* were aligned with those of human, fruit fly, fission yeast and starlet sea anemone. For tractability, we chose not to include plant AGOs. Two prokaryotic AGOs were used to root the tree. A multiple sequence alignment of AGO sequences was constructed using MUSCLE [22]. The phylogenetic tree was constructed with ClustalW [23], excluding positions with gaps and represented with NJPlot [24]. Numbers indicate bootstrapping values. The sequences and the alignment are provided as Supplementary Files. AGOs are represented by a two letter prefix indicating the species, followed by the AGO name or UniProt ID. Species legend: Ce, *Caenorhabditis elegans*; Dm, *Drosophila melanogaster*; Hs, *Homo sapiens*; Mp, *Marinitoga piezophila*; Nv, *Nematostella vectensis*; Rs, *Rhodobacter sphaeroides*; Sp, *Schizosaccharomyces pombe*.

**Figure 2 ncrna-05-00008-f002:**
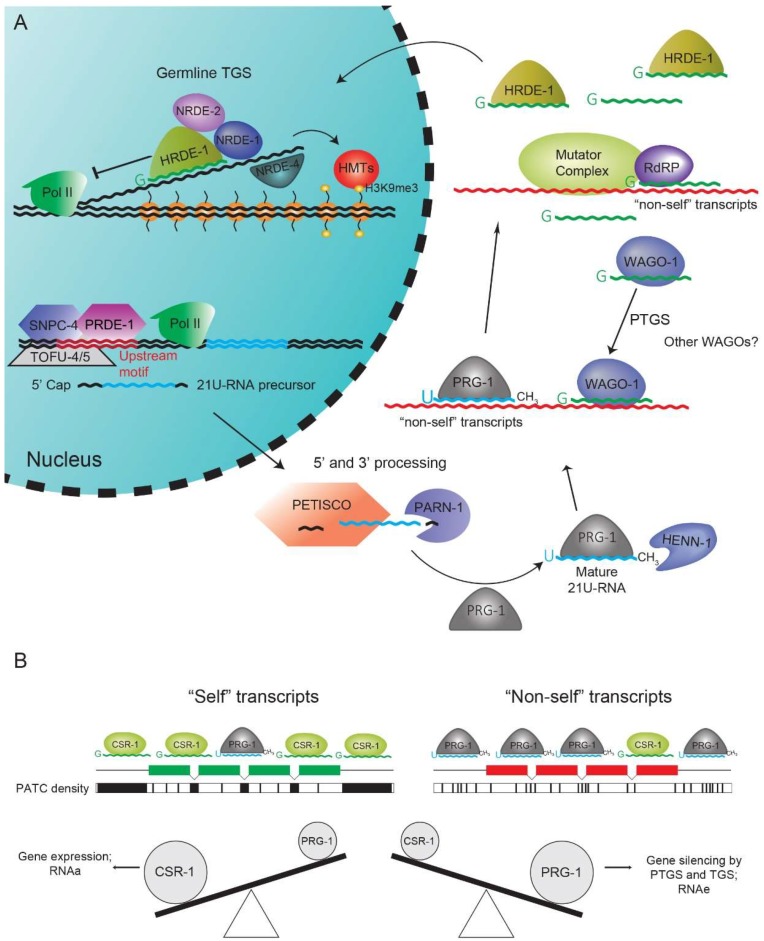
Biogenesis and function of the PRG-1/21U-RNA pathway and its antagonistic signals. (**A**) In the *C. elegans* germline, the USTC complex, with SNPC-4, PRDE-1 and TOFU-4/5, recognize the upstream motif and promote 21U-RNA transcription by RNA PolII. 21U-RNAs are exported to the cytoplasm for further processing. 5′ processing may be undertaken by the PETISCO complex, whereas PARN trims the 3′ ends of 21U-RNAs. 21U-RNAs subsequently associate with PRG-1 and are 2′-O-methylated at the 3′ end by HENN-1. PRG-1 scans the germline transcriptome and stimulates 22G-RNA production, by RdRPs and mutator factors, from foreign transcripts. Then, 22G-RNAs bind to cytoplasmic WAGOs, leading to PTGS of targets or to HRDE-1, which promotes TGS. Accessory NRDE factors are required for HMT recruitment and deposition of repressive chromatin marks. Moreover, NRDE factors may directly impair the activity of RNA PolII. (**B**) CSR-1 and PATCs inhibit repressive PRG-1 signals. The regulatory outcome of a germline-expressed gene will depend on the balance between associated CSR-1/22G-RNAs and PRG-1/21U-RNAs. Germline genes are strongly recognized by CSR-1 and have PATCs throughout their sequence, thereby inhibiting PRG-1 targeting. In contrast, non-self sequences tend to lack the PATC signature and will not be strongly targeted by CSR-1. In this case, stronger PRG-1 targeting will flag these transcripts for degradation. HMT, Histone methyltransferases; PolII, RNA Polymerase II; PTGS, post-transcriptional gene silencing; RdRP, RNA-dependent RNA Polymerase; RNAa, RNA-induced epigenetic gene activation; RNAe, RNA-induced epigenetic silencing; TGS, transcriptional gene silencing.

**Figure 3 ncrna-05-00008-f003:**
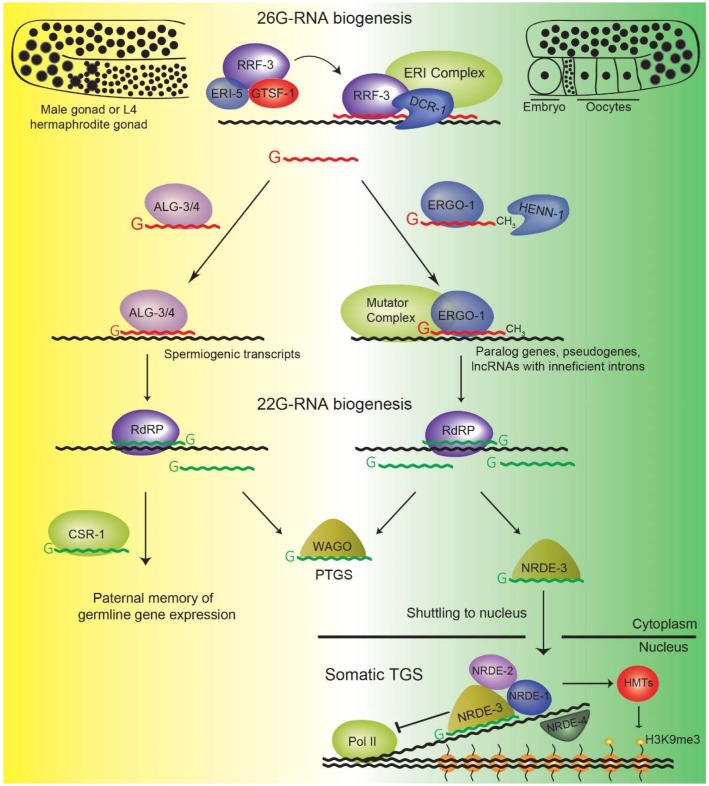
Biogenesis and function of 26G-RNA pathways. Two distinct subpopulations of 26G-RNAs are produced by the ERI complex, which includes the RdRP RRF-3 and DCR-1. To form a mature ERI complex, ERI-5 and GTSF-1 tether RRF-3 to the other ERI complex factors. One subpopulation of 26G-RNAs is produced in the male germline and in the spermatogenic gonad in L4 hermaphrodites (on the left). These 26G-RNAs associate with ALG-3/4 and target spermatogenic transcripts. This is thought to trigger the biogenesis of 22G-RNAs that interact with CSR-1, providing a transgenerational memory of paternal gene expression. In addition, such ALG-3/4-triggered 22G-RNAs may associate with other WAGOs and negatively regulate target gene expression. Another subpopulation of 26G-RNAs is expressed in the oogenic gonad and in embryos (on the right). These associate with ERGO-1 and are 2′-O-methylated by HENN-1 at their 3′ end. ERGO-1-bound 26G-RNAs target a diverse set of genes, which tend to be short, with few introns and with weak splicing signals. ERGO-1 triggers 22G-RNA biogenesis, which in turn can promote PTGS through their association with cytoplasmic WAGOs, as well as TGS through association with the nuclear WAGO protein NRDE-3. NRDE-3, with the assistance of additional NRDE factors and HMTs, perpetuates silencing of 26G-RNA targets in the soma throughout animal development. Of note, NRDE-3 acts in a manner analogous to HRDE-1 in the germline (see Figure 2A). HMTs, Histone methyltransferases; PTGS, post-transcriptional gene silencing; RdRP, RNA-dependent RNA Polymerase; TGS, transcriptional gene silencing.

**Figure 4 ncrna-05-00008-f004:**
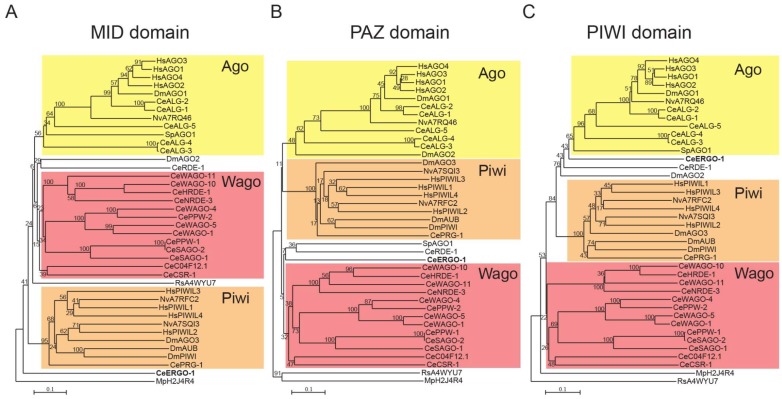
Phylogenetic analysis of the MID, PAZ and PIWI domains of AGO proteins. Phylogenetic trees of the MID (**A**), PAZ (**B**) and PIWI (**C**) domains of AGO proteins. ERGO-1 is represented in bold. The multiple sequence alignment used in Figure 2 was trimmed to the coordinates of either the MID, the PAZ or the PIWI domains, respectively and each tree was constructed as in Figure 2. The alignments are provided as Supplementary Files. AGOs are represented by a two letter prefix indicating the species, followed by the AGO name or UniProt ID. Monophyletic groups including most of the members of one clade are grouped by color. Species legend: Ce, *Caenorhabditis elegans*; Dm, *Drosophila melanogaster*; Hs, *Homo sapiens*; Mp, *Marinitoga piezophila*; Nv, *Nematostella vectensis*; Rs, *Rhodobacter sphaeroides*; Sp, *Schizosaccharomyces pombe*.

**Table 1 ncrna-05-00008-t001:** Conservation of RNAi factors throughout nematodes. Green and gray colored box indicate the existence or absence, respectively, of clear homologs in the designated nematode clade. This table is based on data from refs [107,108,109]. * The Strongyloididae family has lost 5′ triphosphorylated sRNAs.

		Clade I-II	Clade III	Clade IV	Clade V
	DCR-1				
RdRPs	RRF-3				
	EGO-1/RRF-1				
Argonautes	ALG-1/2				
	CSR-1				
	HRDE-1				
	NRDE-3				
	ERGO-1				
	ALG-3				
	PRG-1				
	HENN-1				
sRNA classes	21U-RNA				
	22G-RNA			*	
	26G-RNA				
	miRNA				

**Table 2 ncrna-05-00008-t002:** Comparison between metazoan piRNA pathways and main *C. elegans* RNAi-like pathways. Green and gray colored boxes indicate the existence or absence, respectively, of interaction of the cofactors with the corresponding sRNA class.

		Metazoan piRNAs	21U-RNAs	26G-RNAs
	Expression	Predominantly germline and embryos	Germline and embryos	Germline and embryos
	Length (in nucleotides)	23–35	21	~26
	5′ Bias	U	U	G
	Phenotype	Mutants are sterile	Viable; transgenerational germline mortality	Some mutants cause sterility at higher temperatures
Cofactors	Piwi clade Argonautes				
Hen1 enzymes			
Gtsf1 proteins			
Armitage/MOV1-L1/ERI-6/7

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
