# Peer review of "Function and Evolution of Nematode RNAi Pathways"

_ncrna, 2019, doi:10.3390/ncrna5010008_

Round 1
Reviewer 1 Report
This is a nice review and would make a good addition to the journal, providing it receives a bit more polish. For the most part, the review would provide an excellent introduction and overview to the C. elegans silencing field to new graduate students. Most of the suggestions below are made with that in mind.
· With respect to format, the coloration in the figures is much too dark for printing.
· The paper could use a good proofread using fresh eyes. For example, grammatical mistakes/poor word choice Page 2 Line 76/77, P4L91, P4L137, P6L197, P7L224, P10L315 and others. The legend to Figure 3 needs polish. Also Figure 2: for example “2’-O-methylation at the 3’ end” instead of “3’ 2’-O methyl…” What is meant by “Moreover, the activity of RNA PolII may be thwarted?”
· There are many instances in which the authors have personified their molecular mechanisms, ascribing them emotions such as need, an ability to anticipate, or knowledge of a correct path or outcome. A few examples: P6L174, P6L182, Fig3Legend, P11L378. Such word choice is a reflection of language limitations and usually results in a shorter version of an otherwise much longer description. However consistently thinking in this manner can lead to scientific bias and a failure to correctly interpret data, as we cannot always anticipate what the cell is doing.
· Figure1 could use an explanation regarding why those particular species were selected (and not others).
· There are several places where a brief elaboration would help complete the thought and provide improvement. P4L91 (what machinery?), P4L125 (examples of bottlenecks),
· what is the relative abundance of the sRNA species in C elegans 21U (type I versus II), 22G, 25G? How many 21U genes are in the C. elegans genome?
· “Ruby motif” is jargon to scientists outside the C. elegans RNA field. Its mention should include a brief description. Likewise, many of the acronyms are undefined. A list of acronyms, including gene names, would greatly improve readability for scientists outside the field.
· P6L165, in particular, what abovementioned “new” observations?
· Overall, the paper is lacking in examples of mRNAs that are targeted by the sRNA classes, in particular TE targets, which may help bolster the concluding argument.
· P7L230. It is not apparent how the CSR-1-related data mentioned here, while interesting, is related to temperature sensitivity.
· The 1st paragraph of P9 should be reorganized for clarity.
· On Page 9 Line 272-4, The Eri phenotype is attributed to lack of 22G-RNAs. On P7L208, the Eri phenotype is attributed to shared components in RNAi pathways that are in limited supply.
· The paragraph on P10L344 compares PRG-1 and CSR-1 functions with respect to “transcriptome-wide surveillance” versus “germline-expressed genes”. Isn’t this really just a reflection of where the proteins are expressed?
· Section 8 on P11 regarding cross- and self-regulation is only one paragraph. There is room to expand on the observation that ALG-3/4s are “self-regulating” with respect to implications or consequences to the larger RNAi pathways.
· Tables I and II could be organized a bit better. For example, move the RdRP header to the left so that it is more obviously linking the RdRP genes by function. And make RdRP and Argonaute plural. Why are ego-1 and rrf-1 lumped together? An explanation for how the coloration pattern was determined should be provided. How many species were examined? Which species? Does absence mean complete absence of the gene or limited (%?) conservation, for example?
· P13L455: What form of DNA methylation?
· With respect to the final argument, the authors do not highlight data, or lack thereof, providing evidence of sRNAs targeting TEs in C. elegans.
Author Response
This is a nice review and would make a good addition to the journal, providing it receives a bit more polish. For the most part, the review would provide an excellent introduction and overview to the C. elegans silencing field to new graduate students. Most of the suggestions below are made with that in mind.
· With respect to format, the coloration in the figures is much too dark for printing.
Reply: We thank the reviewer for pointing this out. In our hands this is not a big issue, but we will welcome suggestions from the editors and journal to improve this.
· The paper could use a good proofread using fresh eyes. For example, grammatical mistakes/poor word choice Page 2 Line 76/77, P4L91, P4L137, P6L197, P7L224, P10L315 and others. The legend to Figure 3 needs polish. Also Figure 2: for example “2’-O-methylation at the 3’ end” instead of “3’ 2’-O methyl…” What is meant by “Moreover, the activity of RNA PolII may be thwarted?”
Reply: We have taken into account the suggestions of the reviewer and performed textual changes throughout the manuscript to improve these aspects
· There are many instances in which the authors have personified their molecular mechanisms, ascribing them emotions such as need, an ability to anticipate, or knowledge of a correct path or outcome. A few examples: P6L174, P6L182, Fig3Legend, P11L378. Such word choice is a reflection of language limitations and usually results in a shorter version of an otherwise much longer description. However consistently thinking in this manner can lead to scientific bias and a failure to correctly interpret data, as we cannot always anticipate what the cell is doing.
Reply: We have altered the text accordingly. However, we preferred to maintain personification in certain instances where we believe that scientific correctness is not sacrificed for the sake of clear storytelling. For example, a small RNA “guides” an Argonaute to its target. We think this sort of expression, while personifying small RNAs and Argonautes, still accurately represents molecular events, and is very common in the field.
· Figure1 could use an explanation regarding why those particular species were selected (and not others).
Reply: We have now added the following sentences to the legend of Figure 1: “To provide a broad phylogenetic representation across eukaryotes, the AGO proteins of C. elegans were aligned with those of human, fruit fly, fission yeast and starlet sea anemone. For tractability, we chose not to include plant AGOs.”
· There are several places where a brief elaboration would help complete the thought and provide improvement. P4L91 (what machinery?), P4L125 (examples of bottlenecks),
Reply: We have rewritten those sentences, replacing the term “machinery” in line 93 and elaborating on the bottlenecks in lines 127-128.
· what is the relative abundance of the sRNA species in C elegans 21U (type I versus II), 22G, 25G? How many 21U genes are in the C. elegans genome?
Reply: We added the suggested information where relevant. Please see, for example, lines 134, 192 and 299-300.
· “Ruby motif” is jargon to scientists outside the C. elegans RNA field. Its mention should include a brief description. Likewise, many of the acronyms are undefined. A list of acronyms, including gene names, would greatly improve readability for scientists outside the field.
Reply: We have now eliminated all the references to Ruby motif in text and instead refer to the motif as “upstream motif”. If the editor agrees an extensive list of acronyms (mostly gene names) is needed, we would be happy to provide it.
· P6L165, in particular, what abovementioned “new” observations?
Reply: We altered the sentence to: “These results, together with the observations by Beltran and colleagues [55] indicate…”.
· Overall, the paper is lacking in examples of mRNAs that are targeted by the sRNA classes, in particular TE targets, which may help bolster the concluding argument.
Reply: We address this point in our reply to the last comment of the reviewer. For the purpose of our review, we mention in general terms the sets of targets of each (sub)population of small RNAs. Thus, we believe that providing specific examples goes beyond the scope of our discussion.
· P7L230. It is not apparent how the CSR-1-related data mentioned here, while interesting, is related to temperature sensitivity.
Reply: We have changed this section and think this connection is clearer now.
· The 1st paragraph of P9 should be reorganized for clarity.
Reply: We thank the reviewer for the comment. We have reorganized and rewritten parts of the paragraph as suggested.
· On Page 9 Line 272-4, The Eri phenotype is attributed to lack of 22G-RNAs. On P7L208, the Eri phenotype is attributed to shared components in RNAi pathways that are in limited supply.
Reply: We do not see the problem, as the lack of endogenous 22G-RNAs would free limited shared components for exogenous RNAi. However, for clarity we changed the sentence in lines 274-276 to: “Eri mutants display defects in the accumulation or function of ERGO-1 branch 26G-RNAs and of their respective secondary 22G-RNAs [33,45,71,75,81].”
· The paragraph on P10L344 compares PRG-1 and CSR-1 functions with respect to “transcriptome-wide surveillance” versus “germline-expressed genes”. Isn’t this really just a reflection of where the proteins are expressed?
Reply: The reviewer raised a good point. We changed the sentence as follows, in order to make it clearer: “In the C. elegans germline, PRG-1-RISCs have been shown to perform a transcriptome-wide surveillance of transcripts [35,37,63,64], while CSR-1 targets expressed genes and seems to promote their activity [93–95].”
· Section 8 on P11 regarding cross- and self-regulation is only one paragraph. There is room to expand on the observation that ALG-3/4s are “self-regulating” with respect to implications or consequences to the larger RNAi pathways.
Reply: In principle we agree. However, the finding that ALG-3/4 are engaged in a negative feedback loop is very new (Almeida et al, 2018, BioRxiv). Other than its description, no other experiments were performed to further dissect this regulatory loop and therefore, we prefer not to speculate too far about potential implications.
· Tables I and II could be organized a bit better. For example, move the RdRP header to the left so that it is more obviously linking the RdRP genes by function. And make RdRP and Argonaute plural. Why are ego-1 and rrf-1 lumped together? An explanation for how the coloration pattern was determined should be provided. How many species were examined? Which species? Does absence mean complete absence of the gene or limited (%?) conservation, for example?
Reply: ego-1 and rrf-1 are highly similar RdRPs that work redundantly to some extent (see for example Bagijn et al, 2012). By finding a common proline-rich loop in RRF-1 and EGO-1, but not on RRF-3, Sarkies et al, 2015 refer to an RRF-1/EGO-1 family of RdRPs. We agree and chose to also adopt this grouping throughout our manuscript. In addition, we have reformatted the table as suggested and included a more thorough description of the shading.
· P13L455: What form of DNA methylation?
Reply: In P13L455-456, one can now read “Supporting the latter, RNA-directed DNA methylation…”.
· With respect to the final argument, the authors do not highlight data, or lack thereof, providing evidence of sRNAs targeting TEs in C. elegans.
Reply: We have modified the sentence in lines 182-183 to include the information that TEs are also targeted by 21U-RNAs. In addition, we call to the reviewer’s attention lines 377-379 where we mention TE targeting.
Reviewer 2 Report
This excellent review of nematode small RNAs, from a lab at the forefront of small RNA biology, provides a through overview of the different classes of small RNAs in nematodes while challenging our current classification of C. elegans 21U-RNAs as piRNAs. It makes for an enjoyable and thought-provoking read.
I’m not suggesting any changes but one could argue that the association with Piwi, a central role in transposon silencing, and Dicer-independent biogenesis, clearly place the C. elegans 21U-RNA pathway above the 26G-RNA pathway as the most functionally homologous pathway to piRNAs in flies and vertebrates. From an evolutionary perspective it would seem phylogenetic relatedness points to a common ancestral pathway giving rise to the 21U-RNA pathway in C. elegans and piRNA pathways in other species. Nonetheless, the point that the lines are somewhat blurred and that the standard small RNA classification system is somewhat inaccurate is well taken.
Table 1 shading does not match description. Also, it is not clear what light shading implies.
Table 2. Shading not described.
Typo in lines 508-509. “Are there sRNA features that can help us define “the” worm piRNA pathway? have a distinct 5’ uridine bias which is shared by 21U‐RNAs [8,18,25,36,45].” Some text appears to have been omitted here.
Line 614. “4. Mature piRNAs are consensually….” I don’t think consensually is the right word to use here.
Author Response
This excellent review of nematode small RNAs, from a lab at the forefront of small RNA biology, provides a through overview of the different classes of small RNAs in nematodes while challenging our current classification of C. elegans 21U-RNAs as piRNAs. It makes for an enjoyable and thought-provoking read.
I’m not suggesting any changes but one could argue that the association with Piwi, a central role in transposon silencing, and Dicer-independent biogenesis, clearly place the C. elegans 21U-RNA pathway above the 26G-RNA pathway as the most functionally homologous pathway to piRNAs in flies and vertebrates. From an evolutionary perspective it would seem phylogenetic relatedness points to a common ancestral pathway giving rise to the 21U-RNA pathway in C. elegans and piRNA pathways in other species. Nonetheless, the point that the lines are somewhat blurred and that the standard small RNA classification system is somewhat inaccurate is well taken.
Table 1 shading does not match description. Also, it is not clear what light shading implies.
Reply: We introduced more informative table legends and corrected the shading.
Table 2. Shading not described.
Reply: We have now included the following sentence in the legend of Table 2: “Green and gray colored boxes indicate the existence or absence, respectively, of interaction of the cofactors with the corresponding sRNA class.”
Typo in lines 508-509. “Are there sRNA features that can help us define “the” worm piRNA pathway? have a distinct 5’ uridine bias which is shared by 21U‐RNAs [8,18,25,36,45].” Some text appears to have been omitted here.
Reply: Thank you for pointing it out. We have now corrected it as follows: “…piRNAs have a distinct 5’ uridine bias…”.
Line 614. “4. Mature piRNAs are consensually….” I don’t think consensually is the right word to use here.
Reply: We have replaced “consensually” with “generally”.
Reviewer 3 Report
A very timely review that summarizes the very complex RNAi pathway in nematodes for the general community. I like the authors' approach to bring out the piRNA-like features in the various pathways, highlighting how they all work to combat non-self genetic elements. Will make a very interesting reading for a broad section of the small RNA and transposon community.
Author Response
A very timely review that summarizes the very complex RNAi pathway in nematodes for the general community. I like the authors' approach to bring out the piRNA-like features in the various pathways, highlighting how they all work to combat non-self genetic elements. Will make a very interesting reading for a broad section of the small RNA and transposon community.
Reply: We thank the reviewer for the kind comments.